# Study on the Pb²⁺ Consolidation Mechanism of Gangue-Based Cemented Backfill

**Hao Wang [1,\*]** , **Qi Wang [1]** , **Yuxin Hao [2]** , **Yingying Wang [3]** , **Burui Ta [4]** and **Jian Meng [1]**

[1]    School of Civil Engineering, Zhengzhou University of Technology, Zhengzhou 450044, China
[2]    School of Energy and Mining Engineering, China University of Mining and Technology (Beijing), Beijing 100083, China
[3]    JCHX Mining Management Co., Beijing 100070, China
[4]    Northwest Geological Exploration Institute, China Metallurgical Geology Bureau, Xi'an 710119, China
[\*]    Correspondence: whole_wong@zzut.edu.cn

**Abstract:** Coal mining produces a large amount of gangue that pollutes the environment, causing surface subsidence and damaging the groundwater systems. Backfill mining is an effective technology used to solve this problem, but there is a risk of polluting the groundwater due to the heavy metal ions present in the backfill material. Pb²⁺ has been determined to be a representative element because of its existence in coal gangue samples but not in fly ash. The risk of gangue-based cemented backfill causing groundwater pollution can be evaluated by studying the Pb²⁺ leaching from gangue under various conditions. When comparing the leaching amounts of Pb²⁺ from the coal gangue particles and the test blocks, it was found that cement filling has an obvious consolidation effect on the Pb²⁺ in coal gangue. The above process shows that cemented backfill has an obvious consolidation effect on the Pb²⁺ in gangue. The results of the theoretical analysis, X-ray, and SEM show that the consolidation mechanism can be divided into four modes: physical encapsulation, ion exchange, ion adsorption, and chemical reaction. The results are of great significance for revealing the leaching mechanism of the heavy metals in coal gangue, assessing the risk of heavy metal pollution in groundwater via gangue-cemented backfill, and improving the mining theory of the gangue-cemented filling and groundwater protection.

**Keywords:** gangue; cemented backfill; stress; Pb²⁺; leaching behavior; physical encapsulation; ion exchange; ion adsorption; chemical reaction

## 1. Introduction

While coal mining meets energy demands, gangue covers vast areas of land and pollutes the environment [1,2]. Backfill mining can prevent surface subsidence, maintain buildings and regional water systems, improve the resource recovery rate, and solve the problems of waste accumulation and the waste of land resources, which is in line with the concept of a "green mine" [3–7]. The filling body is buried deep for long periods of time and faces the coupling effect of "force, heat, chemical and seepage". The heavy metal elements carried by gangue leach due to hydrolysis, dissolution, and ion exchange increase the risk of groundwater pollution [8,9].

Reducing the impact of waste on the environment is a relevant issue, and scientists around the world are trying to minimize it. In recent years, a great amount of work has been carried out to study a material created on the basis of enrichment tailings, with the aim of its subsequent use in closed, waste-free (low-waste) production, or in products intended for civil engineering [10,11]. The continuous water tank leaching test developed by the European Union Organization for Standardization shows the leaching characteristics of the hazardous substances of large materials under normal use, and can accurately explain the release mechanism and release behavior of the heavy metals in gangue-based cement filling materials [12,13]. Wang [14] found that the different heavy metal elements in gangue have

different leaching rates, and that the leaching process is long-term. Yao [15] studied the influence of an alternate dry and wet environment, and of the pH of a soaking solution on the hydrogeochemical effect of the heavy metal release in coal gangue with experimental tests. Qi [16], Xu [17], and Song [18,19] studied the relationship between the leaching strength of the heavy metal elements from backfill body and the concentration of those elements in an external solution. Huang [20] et al. found that the leaching strength of the heavy metal elements of weathered gangue is higher than that of those in fresh coal gangue, because of significant changes in its compactness and integrity.

Many scholars have studied the consolidation effect of cementation on heavy metals, such as Liu [21] and Zhao [22], who found that the proportion of impermeable pores in the filling body increases with an increase in the proportion of bonded sand, that the hydration products become more compact and improve the mechanical properties, and that the heavy metal solidification rate exceeds 70%. Shi [23] and Pan [24] characterized the compressive strength and pore structure of backfilling samples with different calcination and binder ratios, and evaluated the leaching concentration of the heavy metals. Liu [25] studied the leaching behavior of the heavy metals in tailings at different stages of the consolidation discharge, and clarified the blocking effect of cementitious materials on the heavy metal migration in the consolidation of tailings. The research results of Liu [26] and others showed that cemented backfill has a solidification effect on the heavy metals in backfill materials, and the ecological risk of using coal gangue as a cemented backfill material is low. Similar conclusions were also proposed in the research results on tailings-cemented backfill [27]. Liu [28] et al. found that the hydration reaction, physical encapsulation, and chemical reaction to the precipitate heavy metal ions play a role in heavy metal consolidation. Santoro [29] and others found that the ettringite and C-S-H gel generated by the hydration reaction can adsorb the heavy metals. Peysson [30] et al. found that the hydration reaction's products exchange ions with the heavy metals to achieve the chemical fixation of the heavy metal ions.

The strength of the backfill body depends on factors such as the component proportion [31,32], the type and dosage of the additives [33–35], the particle size [36,37], and the external environment during the consolidation process [38]. Additionally, the size of the pores and fractures, which are determined by the above factors in the backfill body, can be shown via an X-ray or CT, etc., as can the patency of the leaching channel for the heavy metals [39–41]. Fu [42] and Wu [43] et al. revealed that the failure of the backfilling body includes four stages: microcrack closure, linear elastic, microcrack propagation, and crack penetration failure. Guo [44] found that the failure mode of the gangue-cemented filling material gradually changed from a splitting failure to a shear failure with an increase in size. Liu [45] found that the destruction of the gangue-based cemented backfill mainly occurred at the interface of the coarse aggregate and cementitious material. Zhang [46] found that the permeability of the gangue-based cemented backfilling increases with an increase in the acid corrosion time and osmotic pressure, and decreases with an increase in axial pressure. The greater the degree of the damage to the cemented backfill body, the more developed the internal fractures and the more evident the connection between the fractures are, including the formation of network fractures. The above research mainly focuses on the failure procedure of the backfilling body, which has a significant impact on the leaching channel of the heavy metals, thus affecting the leaching intensity of the heavy metals.

Previously conducted studies have established that the gangue direct backfill pollutes the groundwater due to its presence of heavy metal ions. However, the pollution risk of the gangue-based cemented backfill needs to be studied by considering the change in the gangue particle size during the cemented backfill, as well as the physical and chemical reaction of fine particles such as fly ash and cement during the consolidation process. The purpose of this study is to estimate and reduce the disasters stemming from gangue cemented backfill mining. To achieve this, it is necessary to carry out the following tasks: (1) select appropriate elements and track their leaching behavior; (2) screen for the particle

size and prepare samples, ICP tests, and strength tests, etc.; and (3) conduct an appropriate theoretical analysis.

This paper establishes the relationship between the stress environment and the leaching strength of the heavy metal elements in the backfilling body via experiment tests, and analyzes the consolidation mechanism of cement filling for the heavy metal elements in gangue, providing a reference for mine environmental control.

## 2. Materials and Methods

### 2.1. Pollution Risk of Backfill Mining in Coal Mines

In recent years, based on the increasingly strong demand for environmental protection and improvement in mining, backfill mining has become more popular. Gangue-based cemented backfill mining mixes broken gangue (as coarse aggregate), fly ash (as fine aggregate), and cementitious material to form a slurry with a concentration of 72~78%, and is then transported to the goaf through pipelines to maintain the stability of the rock stratum. The backfill body consolidation with slurry has the characteristics of an early formation strength and a high compressive strength [47–49]. The process of gangue-based cemented backfill mining is shown in Figure 1.

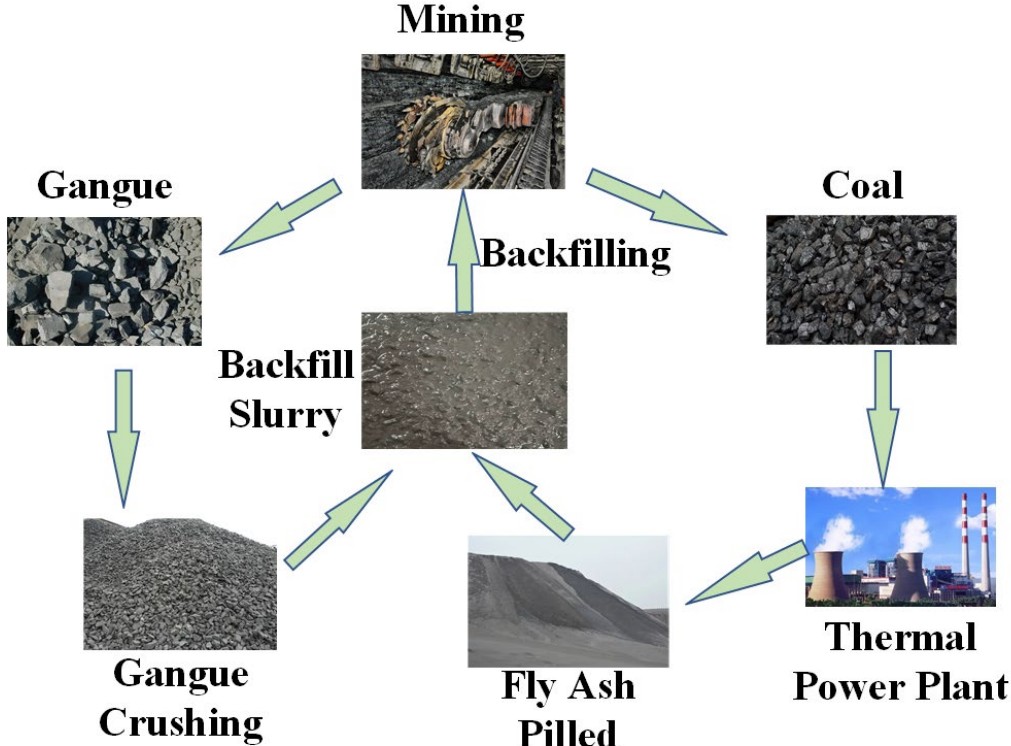

**Figure 1.** Schematic map of backfill in coal mining.

The kind and quantities of the heavy metal elements contained in gangue are different according to their regional geological conditions, but their pollution effect on groundwater systems is essentially the same. The backfill body is buried deeply underground, and its heavy metal elements continuously leach according to many external factors. As the heavy metal elements continuously migrate and accumulate into groundwater and soil systems, the health of the people at the top of the food chain is endangered because of the accumulative effects seen in the animals and plants in the region. Therefore, heavy metal pollution in the regional mine environment has the characteristics of concealment, stability, durability, transferability, irreversibility, and a high hazard.

## 2.2. Material Composition Analysis

The experimental equipment included an X-ray (Seiko Quality Trustworthy, Suzhou, China;), an inductively coupled plasma spectrometer (ICP) (Thermo Fisher Scientific, Waltham, MA, USA; Figure 2), and a pressure servo (Yongce Group Co., Ltd., Shanghai, China). The experimental materials, such as coal gangue, fly ash, and Portland cement (42.5) (produced by Zhengzhou Tianrui Cement Co., Ltd., Zhengzhou, China), were obtained from Zhengzhou, and the experimental water was ordinary distilled water. The experiments were conducted at room temperature. The chemical composition and the content of the coal gangue and fly ash samples that were analyzed by the X-ray and ICP are shown in Tables 1 and 2, respectively. The result shows that both the gangue and fly ash contain a variety of heavy metal elements. However, a high content of Pb was detected in the gangue, but not detected in the fly ash. Therefore, Pb was selected as the representative element to evaluate the consolidation effect of the cemented backfill on the heavy metal elements in gangue with continuous water tank leaching tests, under the premise of the changes according to the time and stress environment.

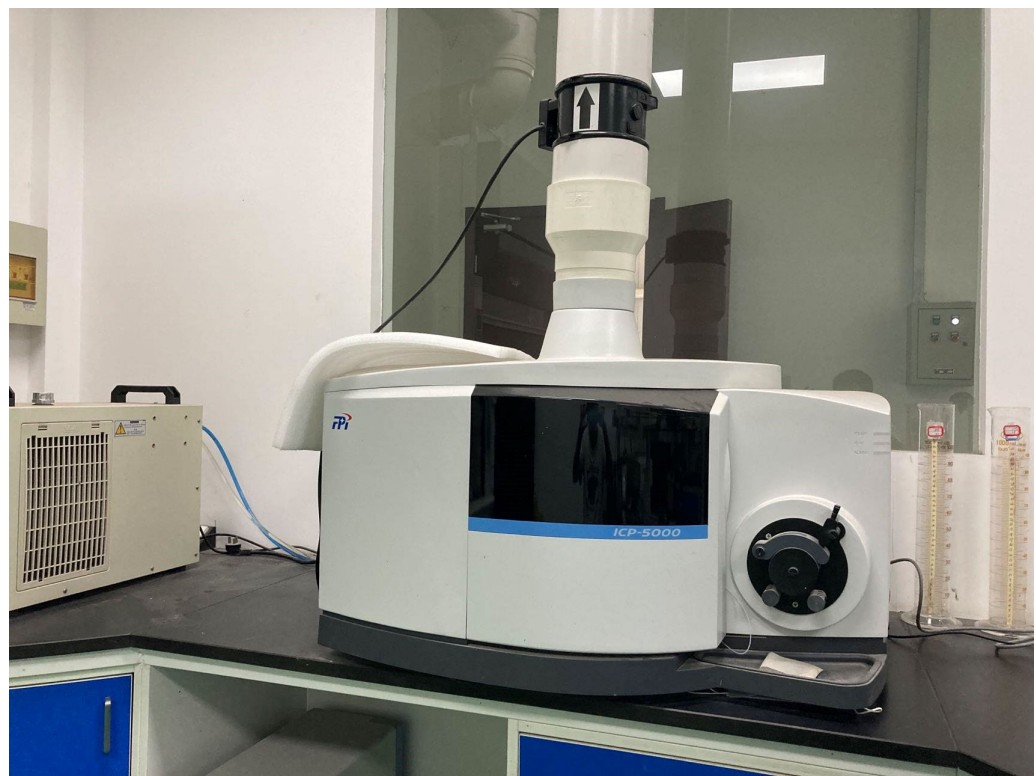

**Figure 2.** Inductively coupled plasma spectrometer.

**Table 1.** Chemical components of fly ash.

| Chemical Composition | SiO$_2$ | Fe$_2$O$_3$ | TiO$_2$ | Al$_2$O$_3$ | CaO | MgO | P$_2$O$_5$ | K$_2$O | LOI |
|---|---|---|---|---|---|---|---|---|---|
| Content (%) | 46.50 | 4.56 | 1.55 | 30.58 | 7.89 | 0.76 | 0.21 | 0.96 | 6.99 |

**Table 2.** Chemical components of coal gangue.

| Chemical Composition | SiO$_2$ | Fe$_2$O$_3$ | PbO | Al$_2$O$_3$ | CaO | MgO | P$_2$O$_5$ | K$_2$O | S | LOI |
|---|---|---|---|---|---|---|---|---|---|---|
| Content (%) | 49.22 | 6.21 | 0.73 | 22.96 | 4.88 | 0.78 | 0.12 | 1.2 | 0.8 | 13.1 |

### 2.3. Strength Test

The strength of the backfill body determines whether the backfill body can have a role in environmental protection. The gangue particles are the skeleton of the backfill body and play an important role in the strength of the backfill body. To highlight the influence of the gangue particle size on the strength of the backfill body and the leaching behavior of $Pb^{2+}$, the gangue was screened according to the particle sizes lower than 0.1 cm, 0.1~0.5 cm, 0.5~1.0 cm, 1.0~1.5 cm, and 1.5~2.0 cm. For convenience of expression, the above-mentioned particle size groups are referred to as 0.1 cm, 0.5 cm, 1.0 cm, 1.5 cm, and 2.0 cm. According to the existing research [50–53], the slurry concentration is 78% and the mass ratio of gangue: fly ash: cement is 8:3:1.

A cement paste mixer was used as the mixing equipment, and the mixing was carried out at a revolution velocity of 62 r/min, and a rotation velocity of 140 r/min; the working time was 18 min. The filling sequence was as follows: all the gangue particles were added to the distilled water, which was added to the mixture of fly ash and cement. It was important to remain vigilant during the periods of high-speed mixing to prevent the water from splashing out from the mixer. Previously, the optimal amount of the Portland cement was determined to be 8% of the solid mass. The Portland cement and the fly ash were mixed for 3 min, then, the distilled water was added, and the mixing was continued for an additional 5 min until a homogeneous mass was achieved. This mixing sequence was the result of a sufficiently small amount of Portland cement, leading to its better distribution within the entire volume of the material being prepared. The strength of the backfill body is a decisive prerequisite for the implementation of backfill mining (T = 20 ± 2 °C; W ≥ 95%). The uniaxial compression test equipment used was YAW-300E, produced by Yongce Group Co., Ltd., Shanghai, China.

The size of sample was 10 cm × 10 cm × 10 cm, and the average value of the 3 samples for each particle size was used. The curves of the compressive strength with the times of the samples, with different particle sizes and times, are shown in Figure 3.

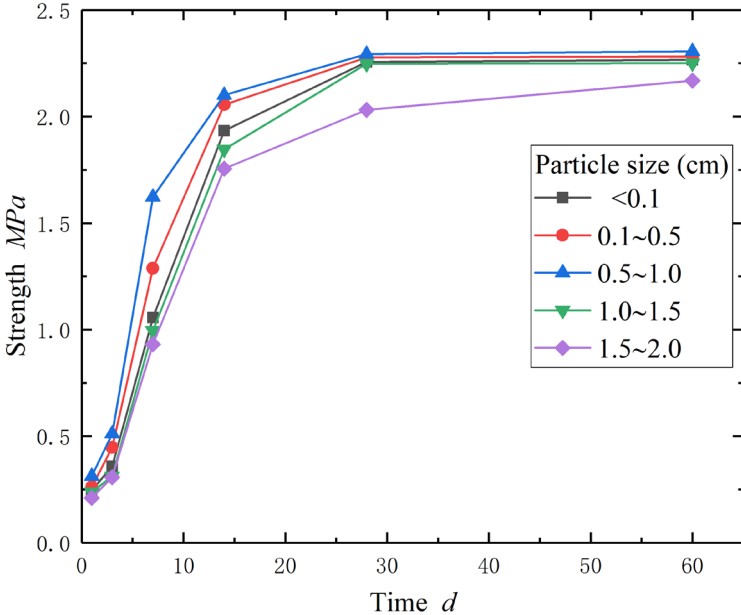

**Figure 3.** Compressive strength versus time curves for samples with different particle sizes.

It can be observed that: the strength of the backfill body increased the fastest in the first 14 days and reached peak strength at day 28, which indicates that a series of physical and chemical changes in the filling material, such as a hydration reaction, mainly occurred in the first 28 days. In this process, fine particles, such as fly ash and cementitious material, formed a cementitious structure with a flocculent network (as shown in Figure 4),

which encloses and consolidates the gangue particles as one, thus showing their obvious strength characteristics.

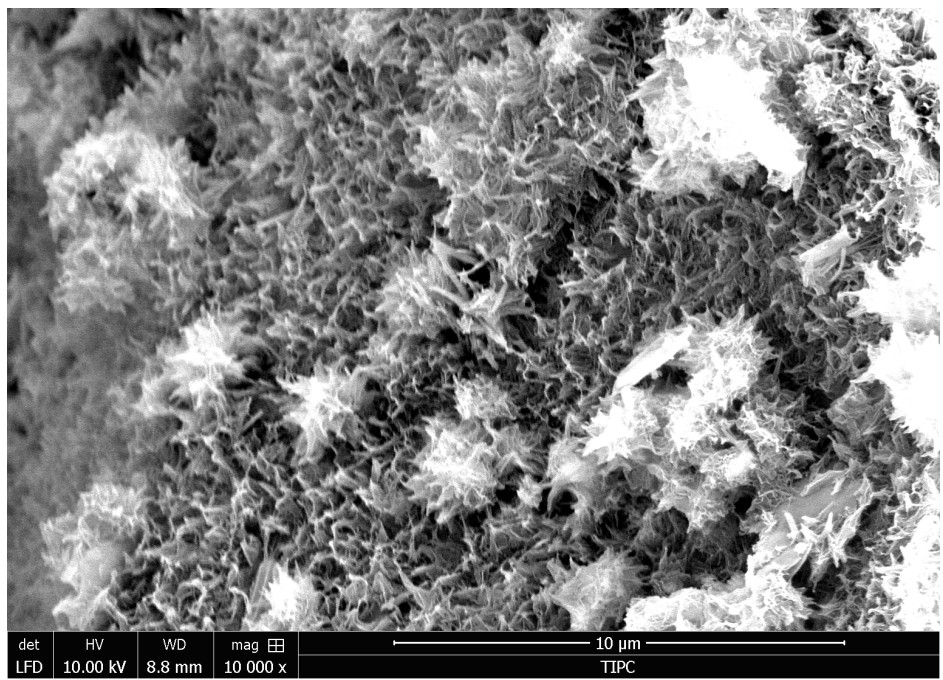

**Figure 4.** Floc structure in block.

Figure 5 shows the strength of the consolidated samples with different particle sizes at day 28. The figure shows that, when the particle size is smaller than 1.0 cm, the strength of the sample increases slightly with an increase in the particle size; when the particle size is greater than 1.0 cm, the strength of the test block decreases with an increase in the particle size. The maximum strength is about 10% higher than the minimum strength.

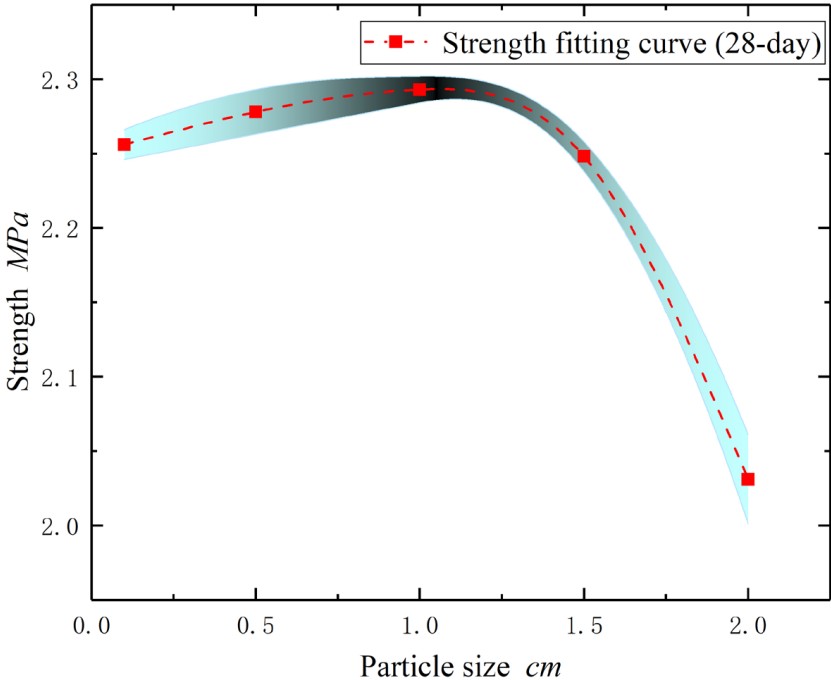

**Figure 5.** Trend curve of samples strength versus particle sizes (28 days).

### 2.4. Pore and Fracture Properties of Filling Body

The surface of the samples at day 28 was relatively rough and full of pores of different sizes and different width cracks, as shown in Figure 6. The mercury intrusion analysis of the samples showed that there was a large number of 0~50 *μm* diameters. The pore diameter conformed to the normal distribution, and the proportion of the 20 *μm* diameter was the largest, as shown in Figure 7. After the backfill body was consolidated, the heavy metal elements in the gangue would leach from the pores and cracks.

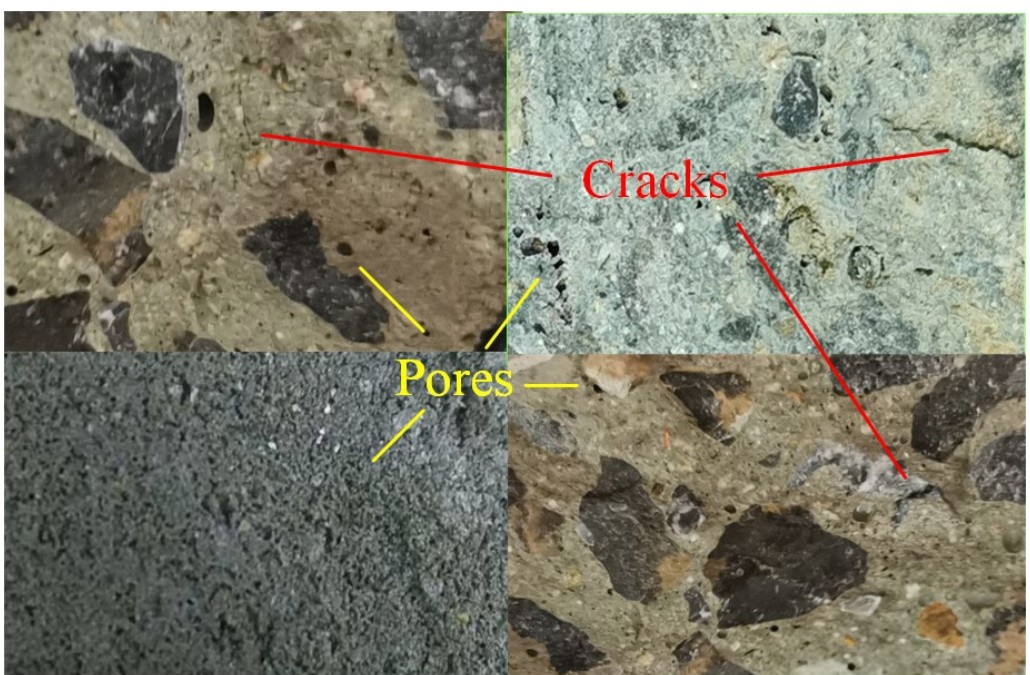

**Figure 6.** Pores and fissures on the samples.

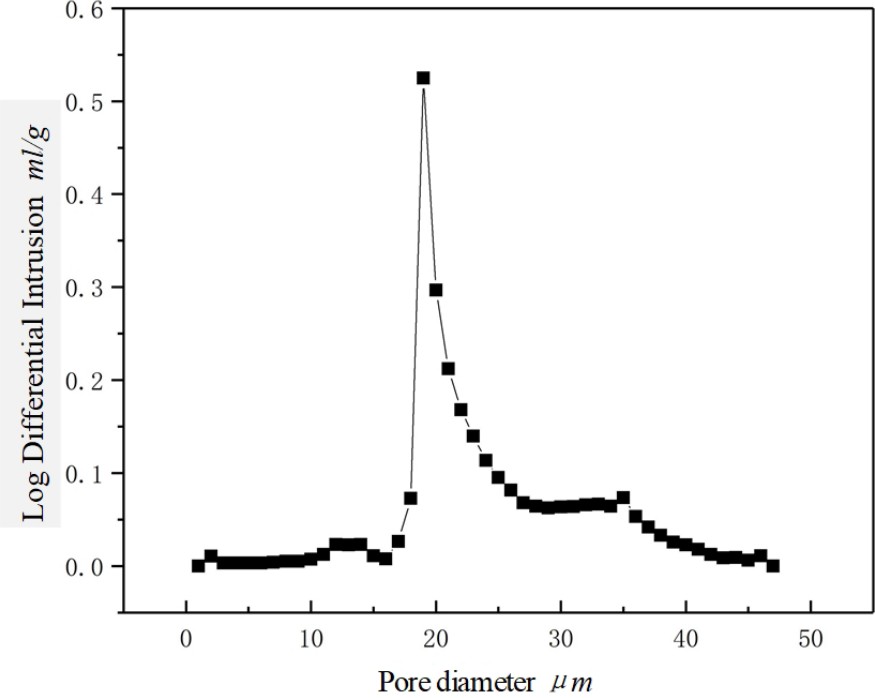

**Figure 7.** Pore diameter distribution of samples (Mercury penetration experiment).

## 3. Results

### 3.1. Leaching Test Process

The solid/liquid ratio was 1:5. The leaching tests were carried out according to Figure 8.

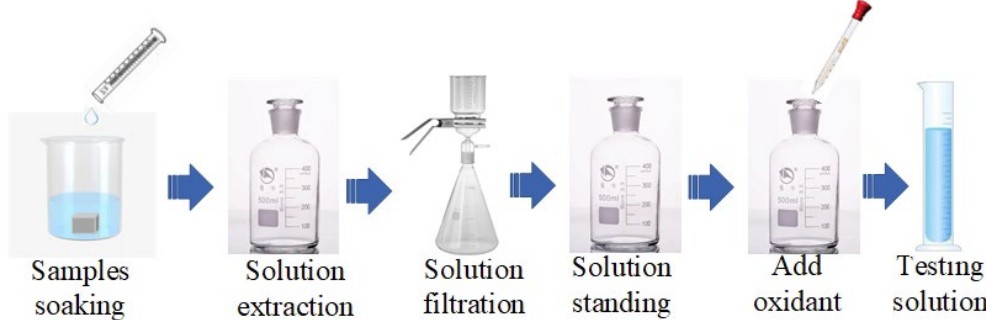

**Figure 8.** Schematic diagram of leaching tests.

The leaching tests belong to the category of the NEN 7375:2004 tank leach test, which is a classic and effective method of evaluating the leaching characteristics and release mechanisms of the hazardous substances in the application scenario of block samples. In the tests, the specimen is subjected to leaching in a closed tank. The leaching solution is renewed after 3, 7, 14, 28, 60, and 120 days. The oxidant is sodium hydroxide solution with a pH = 10.0.

The Pb leaching amount of each stage was calculated according to the formula:

$$E_i = \frac{c_i \cdot V_s}{f \cdot \rho_m \cdot V_m}$$

$E_i$ is the release amount of the Pb in phase $i$, mg/kg; $c_j$ is the mass concentration of the Pb in the leaching solution of phase $i$, $\mu m$; $V_s$ is the leaching solution volume; $f$ is the conversion factor; $\rho_m$ is the sample density; and $V_m$ is the sample volume.

The Pb cumulative leaching amount was calculated according to the formula:

$$\varepsilon_n^* = \sum_{i=1}^{n} E_j$$

$\varepsilon_n^*$ is the Pb cumulative leaching amount.

### 3.2. Effect of Particle Size on the Leaching Strength of $Pb^{2+}$

The object of study of the leaching test was the five groups of the selected coal gangue with different particle sizes. The change in the $Pb^{2+}$ molar concentration in the solution after 3, 7, 14, 28, 60 and 120 days was measured to evaluate the effect of the particle size of the gangue versus the $Pb^{2+}$ leaching strength. The results are shown in Figure 9.

Figure 9 shows that: (1) The $Pb^{2+}$ leaching strength has a significant negative correlation with the particle size of the gangue. The smaller the particle size of the gangue, the greater the $Pb^{2+}$ leaching strength. This means that the smaller the particle size of the gangue, the larger its contact surface with the solution, and the higher the leaching strength of the $Pb^{2+}$ under the same conditions. (2) The leaching strength of the $Pb^{2+}$ in the gangue with different particle sizes is far higher than the standard of groundwater, indicating that the direct backfill of the gangue poses a serious pollution risk to groundwater systems. In fact, this conclusion can be verified by an ICP test of the surface and groundwater samples near the coal mine.

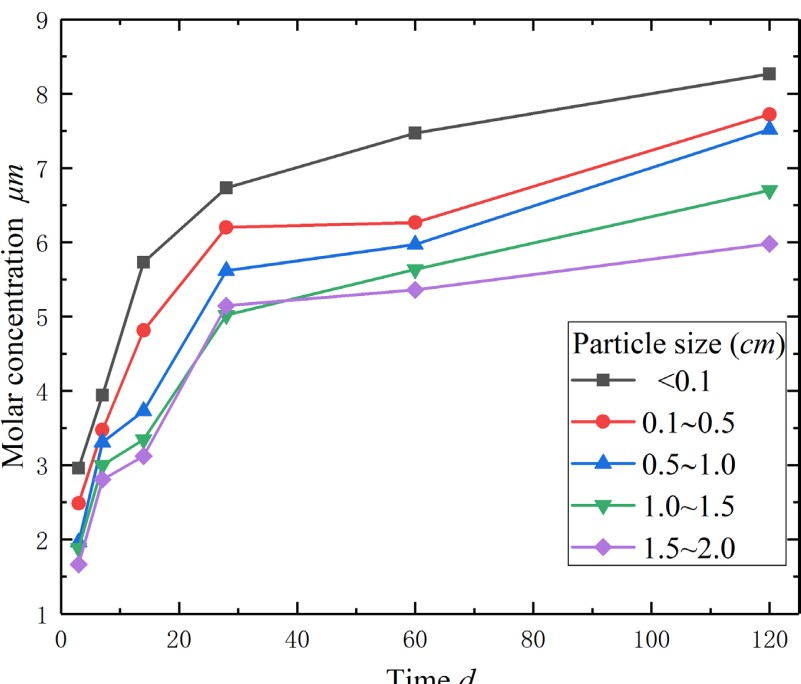

**Figure 9.** Curves demonstrating the amount of $Pb^{2+}$ leaching versus particles sizes.

### 3.3. Effect of Stress on $Pb^{2+}$ Leaching

The gangue-based cemented backfill body is buried deep underground for long periods of time. When the external stress exceeds its ultimate strength, a failure will happen, causing changes to its physical properties. A $Pb^{2+}$ leaching test was conducted using gangue, whose particle size was 1.0 cm in the initial cementation state and under a different external stress. The external stress was set as 0 times, 0.25 times, 0.5 times, 0.75 times, 1 times, and 1.25 times of the strength of the sample, to investigate the effect of both the stress and the sample physical state of the $Pb^{2+}$ leaching. The results are shown in Figure 10.

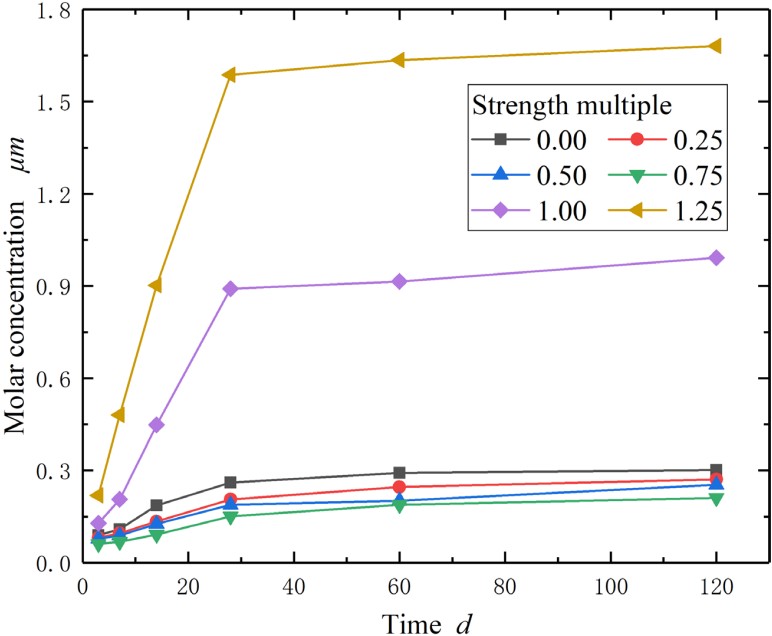

**Figure 10.** Curves demonstrating the amount of $Pb^{2+}$ leaching versus stresses conditions (particle size 1.0 cm).

Figure 10 shows that the $Pb^{2+}$ leaching amount of the gangue in the consolidated state is only about 3% of the rough gangue. The sample leaching amount of the $Pb^{2+}$ in the critical failure state (where the stress is 1.0 times of the strength) is 3~4 times that of the normal state, while the leaching strength of the sample in the full failure state (where the stress is 1.25 times of the strength) is 8~10 times that of the normal state. Even in the full failure state, its $Pb^{2+}$ leaching amount is only about 25% of that of the rough gangue particle. This means that, compared with the direct filling of gangue, the pollution risk of gangue-based cemented backfill to groundwater is significantly restricted.

The leaching amounts of the samples in the first 28 days is large under the different stresses, and then decreases significantly. Especially for the samples in a critical failure state or full failure state, the leaching amount of the $Pb^{2+}$ increases rapidly, more obviously in the first 28 days. The leaching strength of the $Pb^{2+}$ is negatively correlated with external stress when the stress is lower than the ultimate strength, as the leaching strength of the $Pb^{2+}$ decreases with the increase in stress. When the stress of the sample is higher than the ultimate strength, the leaching strength of the $Pb^{2+}$ increases rapidly.

## 4. Discussions

### 4.1. Encapsulation

The fine aggregate (cement and fly ash) undergoes a hydration reaction in the process of consolidation, producing a large amount of granular or fibrous C-S-H ($mCaOnSiO_2 \cdot xH_2O$) gel. The scale of the C-S-H gel increases with time, because the hydration reaction is increasingly more sufficient. This C-S-H gel covers the surface of the gangue particles, greatly reducing their contact area with the external solution. The leaching of the heavy metal elements in the gangue particles is, therefore, limited because of physical encapsulation. Figure 11a shows the samples prepared for the SEM test. Figure 11b shows the microstructure analysis of the cemented backfill body using SEM. It shows that a large amount of the C-S-H gel covers the surface of the gangue particles.

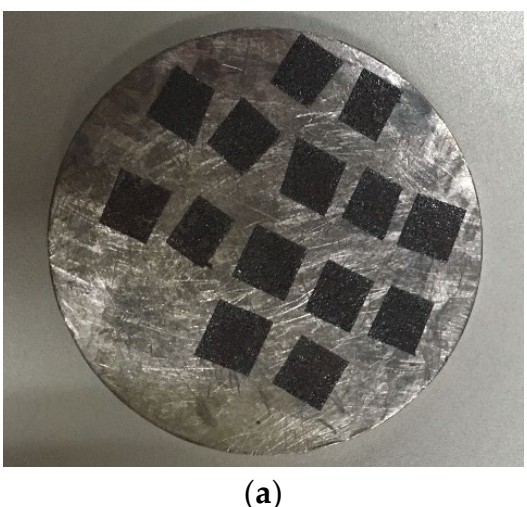
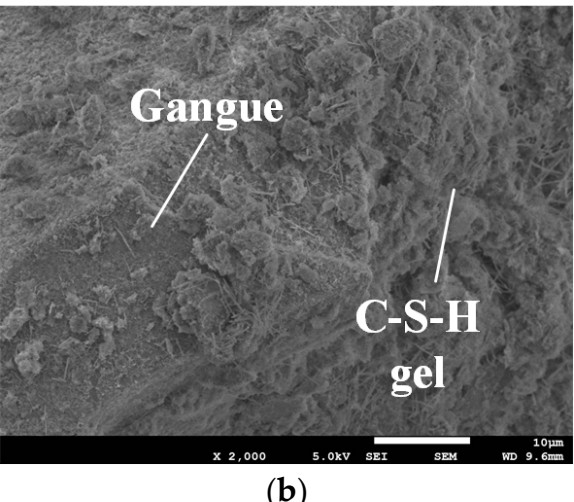

(**a**)　　　　　　　　　　　　　　　　　　(**b**)

**Figure 11.** (**a**) Samples used for SEM; and (**b**) C-S-H gel covers the gangue particles.

### 4.2. Ion Exchange between Hydration Products and $Pb^{2+}$

The X-ray information and the test setup are shown in Table 3.

Figure 12 shows the X-ray analysis of the gangue-based cemented backfill body. The results show that the cement backfill contains quartz, kaolinite, illite, dihydrate gypsum, smectite, mix salt, calcite, and other minerals. The ion exchange process is shown in Equation (1).

$$Pb^{2+} + Ca(C\text{-}S\text{-}H)_2 \rightarrow Pb(C\text{-}S\text{-}H)_2 + Ca^{2+} \tag{1}$$

**Table 3.** The information of the X-ray and the test setup.

| Name | D/Max-3B X-ray diffractometer |
|---|---|
| Manufacturer | Rigaku (Japanese) |
| Test conditions | Cu target, K radiation, graphite curved crystal monochromator |
| Slit system | DS (divergent slit): 1°<br>RS (receiving slit): 1°<br>SS (anti-scattering slit): 0.15 mm<br>RSM (monochromator slit): 0.6° |
| X-ray tube voltage | 35 kV |
| X-ray tube current | 30 mA |
| Qualitative analysis | Scanning mode: continuous scanning;<br>Scanning speed: 3°/min;<br>Sampling interval: 0.02° |
| Quantitative analysis | Scanning mode: step scanning;<br>Scanning speed: 0.25°/min;<br>Sampling interval: 0.01° |
| Databases | The standard powder diffraction data provided by JCPDS-ICDD |

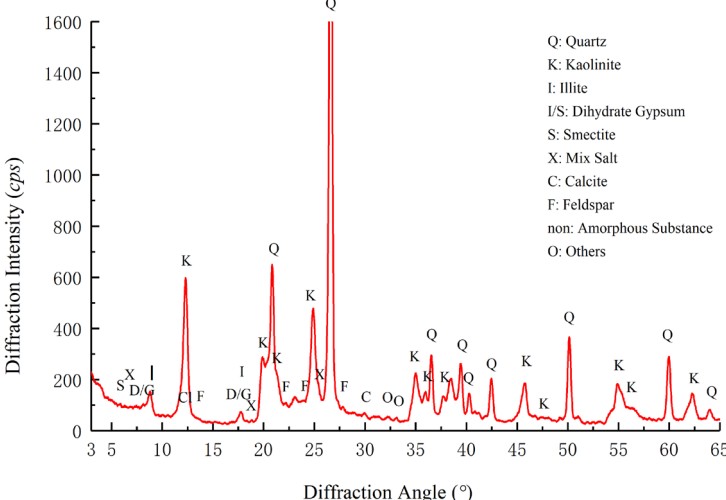

**Figure 12.** X-ray analysis of the samples.

*4.3. Adsorption of Hydration Products on $Pb^{2+}$*

The $Pb^{2+}$ is physically adsorbed by the negatively charged C-S-H. The chemical adsorption of the $Pb^{2+}$ by the C-S-H gel is usually shown in the $Pb^{2+}$ bonds with $Ca^{2+}$ and $Si^{4+}$, and forms precipitates such as mixed salts. The reaction can be verified by the X-ray analysis of the cement backfill.

$$Pb^{2+} + 2(C\text{-}S\text{-}H)^- \rightarrow Pb\text{-}(C\text{-}S\text{-}H)_2 \tag{2}$$

$$3Pb^{2+} + 7OH^- + H^+ + Ca^{2+} + SO_4^{2-} \rightarrow Pb_2SO_4(OH)_2 \cdot Pb(OH)_2 \cdot Ca(OH)_2 \cdot H_2O \tag{3}$$

*4.4. $Pb^{2+}$ Chemical Reaction*

During the consolidation of the filling body, the $Pb^{2+}$ in the gangue participates in the physical and chemical reaction, generating precipitation such as ash and resulting in the $Pb^{2+}$ consolidation.

1.  The reaction between CaO and water, as well as the neutralization reaction between $H^+$ and $OH^-$, and the combination of the $Pb^{2+}$ and $OH^-$ to form precipitated $Pb(OH)_2$, are expressed as:

$$CaO + H_2O \rightarrow Ca(OH)_2 \tag{4}$$

$$H^+ + OH^- \rightarrow H_2O \tag{5}$$

$$Pb^{2+} + OH^- \rightarrow Pb(OH)_2 \tag{6}$$

2. The $SO_4^{2-}$, the external environment reacts with $Ca(OH)_2$, $CaO$, and $Al_2O_3$, and generates hydrated calcium silicate (Equation (7)) and gypsum dihydrate (Equation (8)). It also produces a combination of $Pb^{2+}$ and $SO_4^{2-}$ to produce $PbSO_4$, which is insoluble in water and acid solution.

3. $S^{6+}$ and $O^{2-}$ are combined to generate $SO_4^{2-}$, which, combined with $Ca^{2+}$, can also produce gypsum dihydrate (Equation (8)).

$$Ca^{2+} + SO_4^{2-} \rightarrow CaSO_4 \tag{7}$$

$$4CaO{\cdot}Al_2O_3{\cdot}19H_2O + 2Ca(OH)_2 + 3SO_4^{2-} + 14H_2O \rightarrow 3CaO{\cdot}Al_2O_3{\cdot}3CaSO_4{\cdot}32H_2O + OH^- \tag{8}$$

$$CaSO_4 + 2H_2O \rightarrow CaSO_4{\cdot}2H_2O \tag{9}$$

$$Pb^{2+} + SO_4^{2-} \rightarrow PbSO_4 \tag{10}$$

4. The hydration reaction generates hydrated calcium silicate gel.

$$2(2CaO{\cdot}SiO_2) + 4H_2O \rightarrow 3CaO{\cdot}3SiO_2{\cdot}3H_2O + Ca(OH)_2 \tag{11}$$

5. The $OH^-$ and $H^+$ in the solution enter the filling body, decomposing the ettringite and hydrated calcium silicate gel.

$$3CaO{\cdot}Al_2O_3{\cdot}3CaSO_4{\cdot}32H_2O + OH^- \rightarrow 6Ca^{2+} + 2Al(OH)_4^- + 3SO_4^{2-} + 4OH^- + 26H_2O \tag{12}$$

$$3CaO{\cdot}2SiO_2{\cdot}3H_2 + H^+ \rightarrow 3Ca^{2+} + 2SiO_2 + H_2 \tag{13}$$

It should be noted that the above chemical reactions are not limited to a specific period, and are cycled during the entire process. The reaction intensity changes with the external temperature, stress state, and other factors.

### 4.5. Summary of the Consolidation Mode

The heavy metal consolidation of the cement backfill can be divided into four approaches: physical encapsulation, ion exchange, ion adsorption, and chemical reaction, as shown in Figure 13.

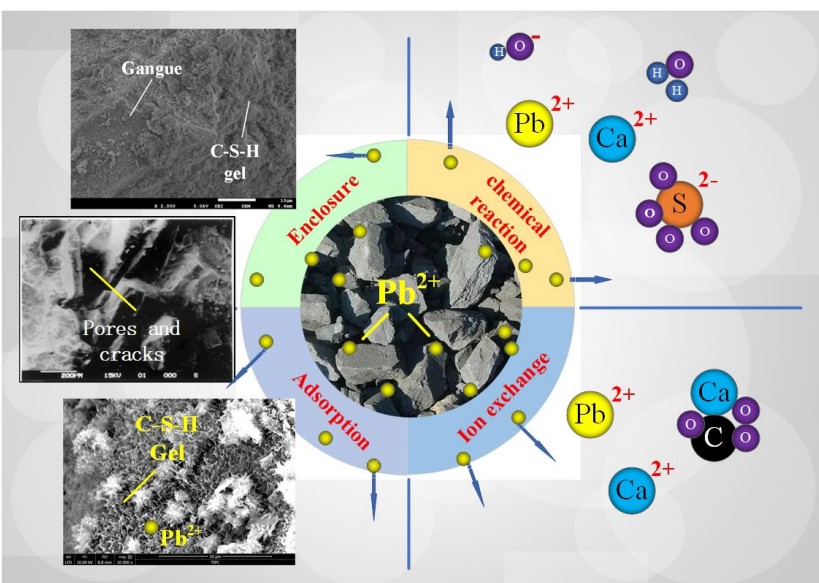

**Figure 13.** $Pb^{2+}$ consolidation mechanism of gangue based cemented backfill.

As the gangue-based cemented backfill body suffers the coupling effect of multiple forces, such as "force, chemistry, heat, seepage", the manner of the heavy metal consolidation will vary with time and the external environment, and with mutual transformation and coordination. For example, the chemical reaction and ion exchange are the main characteristics within the first 28 days of consolidation, because the strength of the filling body is low, and the external solution can enter the filling body in a relatively smooth way, reacting with the heavy metal elements and removing the heavy metal ions. After 28 days of the consolidation of the backfill body, its strength is close to peak values, and the hydration reaction forms a large amount of C-S-H gel, which forms an inclusion for the gangue particles, and the adsorption effect of the heavy metal elements is greatly enhanced. When the filling body is damaged, more gangue particles are exposed and in contact with the solution. At the same time, when the C-S-H gel is damaged, the adsorption effect is relatively reduced, resulting in a greater leaching intensity of the heavy metal elements. The process is also consistent with the leaching curve.

## 5. Conclusions

The paper analyzed the $Pb^{2+}$ leaching strength and the reasons for this occurrence in gangue-based cemented backfill according to different factors, so as to provide support for the prevention of the pollution risks of coal power production and to improve the relevant theory of a "green mine".

(1) The direct backfill of gangue has a high pollution risk with regards to groundwater. According to the leaching curve of the $Pb^{2+}$ of gangue particles (Figure 8), there is a negative correlation between the particle size of gangue and the $Pb^{2+}$ leaching strength, that is, the smaller the particle size, the higher the $Pb^{2+}$ leaching strength. The $Pb^{2+}$ leaching direction gradually extends from the surface of the coal gangue to the interior of the coal gangue along the micro-pores with time, but the $Pb^{2+}$ leaching is only limited near the gangue surface because of its limited size and lack of complete connection to the pores.

(2) Cemented filling has an obvious "consolidation" effect on the $Pb^{2+}$ in gangue, which can greatly reduce the risk of groundwater pollution from the direct backfill of gangue. The influencing factors are: (a) The gangue particles are wrapped by fine particles such as fly ash, greatly reducing the contact surface with the solution, as well as the $Pb^{2+}$ leaching strength. (b) The $Pb^{2+}$ participates in the chemical reaction and forms new macromolecular compounds. (c) The C-S-H gel blocks the $Pb^{2+}$. However, the leaching strength of the $Pb^{2+}$ in the gangue-based cemented backfill was relatively high in the first 28 days, resulting in a loss of groundwater function. Especially if the backfill body is damaged, the leaching amount of the $Pb^{2+}$ is greatly increased. Therefore, the strength of the gangue-based backfill body should be guaranteed to avoid the heavy metal pollution of groundwater systems.

(3) In the $Pb^{2+}$ leaching channel, there are vast quantities of pores and cracks in the cemented backfill body, and their quantity will change according to external stress. When the stress on the samples does not reach the ultimate strength, the pores and cracks are compressed, the contact interface between the gangue and solution is reduced, the $Pb^{2+}$ leaching channel becomes smaller, and the $Pb^{2+}$ leaching strength is reduced. When the stress on the samples reaches or exceeds the ultimate strength, the samples are destroyed and cracks will develop rapidly, not only becoming larger in size but even forming linear or reticular cracks. On the one hand, the amount of coal gangue exposed to the solution increases; on the other hand, the $Pb^{2+}$ leaching channel is obviously smooth.

(4) These conclusions can aid in the prevention and control of the heavy metal pollution of non-metallic mines and other underground landfills.

**Author Contributions:** Conceptualization, H.W. and Q.W.; methodology, H.W.; software, H.W.; validation, H.W. and Q.W.; formal analysis, Q.W., Y.W., B.T. and J.M.; investigation, Q.W.; resources, H.W. and Y.W.; data curation, Y.H., B.T. and J.M.; writing—original draft preparation, H.W.; writing—review and editing, H.W., Y.H., B.T. and J.M.; visualization, H.W. and Y.H.; supervision, Q.W. and Y.W.; project administration, Q.W. and Y.W.; funding acquisition, H.W. All authors have read and agreed to the published version of the manuscript.

**Funding:** This study received financial support from the Funding: Foundation of He'nan Science and Technology Committee (NO. 222102320447).

**Data Availability Statement:** The datasets used and/or analyzed during the current study are available from the corresponding author upon reasonable request.

**Conflicts of Interest:** The authors declare no competing interest.

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
