# Peer review of "Study on the Pb2+ Consolidation Mechanism of Gangue-Based Cemented Backfill"

_minerals, doi:10.3390/min13030354_

Round 1

Reviewer 1 Report

The manuscript describes the experimental results of Pb2+ leaching tests under different conditions. The results are interesting and providing useful information for the literature. However, major improvement of the paper is needed to be a good credible paper. Some specific comments are listed as follows:
1.    The language of the abstract needs to be revised. The same comment is applicable for the Introduction.
2.    In abstract, line 14-15, the sentence “In order to …….heavy metal cement” is not clear to the reader.
3.    How did you determine the chemical composition of fly ash and Coal Gangue as shown in Table 1 and 2?
4.    The experimental procedure of leaching tests are absent in the manuscript. It is important to describe the experimental procedure in details with figure of real test for a credible paper. The test standard with reference should also be included.
5.    Page 5, line 161-163, generally smaller particle is giving higher strength, why in your case it is less? Please explain.
6.    Page 10, line 250, What is Th2.SO42+ ?
7.    Page 10, line 255, the equation 10 should be renumbered?
8.    Section 5 “Results and Discussion” becomes extended conclusions rather discussion.
9.    The conclusions section is missing in the manuscript.
10.    Line 119, Formatting issue. “2.2. Material composition analysis” will be in next line.
11.    Line 193, Formatting issue. “3.2. Effect of stress on Pb2+ leaching” will be in next line.
12.    Overall, the information of the paper needs to be rearranged. For example under “2. Materials and Methods “, you described the results of the test (Fig. 3 to 7). These should be under results and discussion section.

Author Response

Response to Reviewer 1 Comments

Point 1: The language of the abstract needs to be revised. The same comment is applicable for the Introduction.

Response 1: We are very sorry for the poor linguistic quality of this paper. We have invited a professional scholar to polish this paper and improve the linguistic quality. These improvements are written in red colour in the text. We have listed some improvements as follows.

1) Page 1, line 42-line 46:

The sentences were revised as “Reducing the impact of waste on the environment is a relevant issue, and scientists around the world are trying to minimize it. In recent years, a great amount of work has been carried to study a material created on the basis of enrichment tailings with the aim of their subsequent use in a closed, waste-free (low-waste) production or in products intended for civil engineering [10,11].”

2) Page 2, line 91:

The word ”And” was revised as”Additionally”.

3) Page 3, line 102-line 105:

The sentences were revised as “The greater the degree of the damage to the cemented backfill body damage, the more developed the internal fractures and the more evident the connection between the fractures, including the formation of network fractures.”

4) Page 3, line 110-line 119:

We add some sentences to make the make the article more integrity.

“Previously conducted studies have established that gangue direct backfill pollutes the groundwater due to the presence of heavy metal ions. However, the pollution risk of gangue-based cemented backfill needs to be studied by considering the change in the gangue particle size during cemented backfill as well as the physical and chemical reaction of fine particles such as fly ash and cement during the consolidation process. The purpose of this study is to estimate and reduce the disasters stemming from gangue cemented backfill mining. To achieve this, it is necessary to carry out the following tasks: 1) Select appropriate elements and track their leaching behavior; 2) Screen for particle size and prepare samples, ICP tests, strength tests, etc; 3) Conduct an appropriate theoretical analysis.

5) Page 6, line 210:

“a large amount of” was revised as “a large number of”.

And also the other sentences has been revised:

Page 5, line 186:

The sentence was revised as “The curves of the compressive strength with time of samples with different particle sizes and time are shown in Figure 3.”

Page 11, line 291:

The sentence was revised as “The Pb2+ is physically adsorbed by the negatively charged C-S-H.”

Point 2: In abstract, line 14-15, the sentence “In order to …….heavy metal cement” is not clear to the reader.

Response 2: I'm sorry for the ambiguity expression. The sentence “In order to …….heavy metal cement” revised as “Pb2+ determined as the representative element because of it exists in coal gangue samples but not in fly ash. Thethe risk of gangue based cemented backfill with regard to groundwater pollution is evaluated by studying the Pb2+ leaching from gangue under various conditions.” (Page 1, line 16-19)

Point 3: How did you determine the chemical composition of fly ash and Coal Gangue as shown in Table 1 and 2?

Response 3: I am sorry for the missing informations. The numbers means chemical content, we add “%” in the Table 1 and Table 2.

Point 4: The experimental procedure of leaching tests are absent in the manuscript. It is important to describe the experimental procedure in details with figure of real test for a credible paper. The test standard with reference should also be included.

Response 4: I am sorry for the incomplete provide of information.

After discussion carefully, we decided to use the following diagram to deduce the experimentals and the collection of liquid samples. I hope Figure 1 will get the affirmation and support from reviewers.

(Page 7, line 219-line 222)

Figure 8. Schematic diagram of leaching tests

Point 5: Page 5, line 161-163, generally smaller particle is giving higher strength, why in your case it is less? Please explain.

Response 5: In the research, the strength fitting curve (28-day) shows increasing slightly when the particle size is smaller than 1.0cm and rapid decreasing when the particle size is greater than 1.0 cm. The strength of 2.0cm represented sample is about 10% lower than the strength of 0.1cm represented sample. As the reviewer says, “generally smaller particle is giving higher strength”.

Point 6: Page 10, line 250, What is Th2.SO42+ ?

Response 6: I am sorry for the wrong spelling. The “Th2.SO42+” should be revised as “The SO2- 4”. And, thanks for your work on the preciseness of the article.

Point 7: Page 10, line 255, the equation 10 should be renumbered?
Response 7: I am sorry for the wrong number. The equations (Page 11-Page 12, line 307-line 308) were renumbered, also the quotes (Page 11, line302 -line 307). And, thanks for your work on the preciseness of the article.

Point 8: Section 5 “Results and Discussion” becomes extended conclusions rather discussion.
Response 8: Section 5 “Results and Discussion” revised as “5. Conclusions” in order to improve the integrity of the article. And we hope that the reviewers agree with and support the views.

Point 9: The conclusions section is missing in the manuscript.
Response 9: After the information of the paper rearranged, the “5. Results and Discussion” in the original paper revised as “5. Conclusions”.

Point 10: Line 119, Formatting issue. “2.2. Material composition analysis” will be in next line.
Response 10: I am sorry for the wrong format. The “2.2. Material composition analysis” has been put  in next line and revised as “2.2. Material composition analysis”.

Point 11: Line 193, Formatting issue. “3.2. Effect of stress on Pb2+ leaching” will be in next line.
Response 11: I am sorry for the wrong format. The “3.2. Effect of stress on Pb2+ leaching” has been put in next line and revised as “3.2. Effect of stress on Pb2+ leaching.

Point 12: Overall, the information of the paper needs to be rearranged. For example under “2. Materials and Methods “, you described the results of the test (Fig. 3 to 7). These should be under results and discussion section.
Response 12: Thanks for the reviewer’s professional comments. The sections of the paper have been rearranged “2. Materials and Methods”,” 3. Results”,” 4. Discussionsand “5. Conclusions”.

The results of the test in “2. Materials and Methods“ include “Material composition analysis”,” Strength test”,” Floc structure (SEM)”,” Pore diameter distribution”. These tests and results are the premise for the core experiment and results of the article, that is, the consolidation effect of cemented backfill on Pb2+ under different states and influence factors.

Based on the considerations, we think the contents should be placed in “2. Materials and Methods“. And we hope that the reviewers agree with and support the views.

Reviewer 2 Report

The authors are concerned with the leaching and consolidation behaviors of lead ions (Pb2+) during the mine backfilling process using coal gangue and fly ash as filling materials. While the topic is most relevant to Minerals and the introduction is mostly informative, the reviewer would like to reject this paper because it has some serious flaws.

1.       The authors proposed four mechanisms to explain the consolidation behavior of Pb2+ in Section 4, while these mechanisms are not supported by the evidence shown in this paper. For example, in Section 4.1, the authors tried to use one SEM image (Fig. 10) to justify the encapsulation mechanism, i.e., formation of C-S-H gel on the gangue particle surface should inhibit releasing of Pb2+ from gangue particles. This SEM image is poor in quality. The boundary between C-S-H gel and gangue cannot be clearly identified. How did the authors identify the gangue and C-S-H gel in this image since they looked very similar? How did the authors prove that C-S-H gel were formed and existed on the gangue particle surface? Without answering these questions, it is hard to convince the readers with this encapsulation mechanism.

The same problem occurs in Section 4.2. The authors showed the XRD results in Fig. 11 and proposed the ion-exchange theory. The XRD results showed the mineralogy information of the backfill sample, but they did not imply that the ion-exchange reaction for Pb2+. It made no sense to use XRD results to suggest the ion-exchange mechanism.

For the adsorption of Pb2+ on C-S-H mechanism proposed in Section 4.3, Eqs. (2) and (3) were not right. In Eq. (2), the lead ion should have 2 positive charges. In Eq. (3), calcium ions and sulfate ions cannot be present in the aqueous phase at the same time because calcium sulfate will precipitate immediately. Due to these flaws, it is hard to proof the adsorption mechanism.

2.       Some key information about the leaching tests is not given in this manuscript. For example, what is the solid/liquid ratio? What is the pH of leachate? How did the authors collect the liquid sample? Any mixing applied during the leaching tests? What is the device used for the leaching tests? Without these parameters, it is hard to understand the results shown in Figs. 8 and 9.

Author Response

Response to Reviewer 2 Comments

Point 1:  â‘ The authors proposed four mechanisms to explain the consolidation behavior of Pb2+ in Section 4, while these mechanisms are not supported by the evidence shown in this paper. For example, in Section 4.1, the authors tried to use one SEM image (Fig. 10) to justify the encapsulation mechanism, i.e., formation of C-S-H gel on the gangue particle surface should inhibit releasing of Pb2+ from gangue particles. This SEM image is poor in quality. The boundary between C-S-H gel and gangue cannot be clearly identified. How did the authors identify the gangue and C-S-H gel in this image since they looked very similar? How did the authors prove that C-S-H gel were formed and existed on the gangue particle surface? Without answering these questions, it is hard to convince the readers with this encapsulation mechanism.

â‘¡The same problem occurs in Section 4.2. The authors showed the XRD results in Fig. 11 and proposed the ion-exchange theory. The XRD results showed the mineralogy information of the backfill sample, but they did not imply that the ion-exchange reaction for Pb2+. It made no sense to use XRD results to suggest the ion-exchange mechanism.

â‘¢For the adsorption of Pb2+ on C-S-H mechanism proposed in Section 4.3, Eqs. (2) and (3) were not right. In Eq. (2), the lead ion should have 2 positive charges. In Eq. (3), calcium ions and sulfate ions cannot be present in the aqueous phase at the same time because calcium sulfate will precipitate immediately. Due to these flaws, it is hard to proof the adsorption mechanism.

Response:

â‘ We regret that we have not fully explained this part.

The particle size of gangue in the sample used in the SEM experiment is far less than 0.1cm. Even so, it is still unable to completely present its particle appearance in the experiment because it is magnified more than 2000 times (Figure 10(a)).

Figure 10(a). Samples used for SEM

In order to more clearly show the structure and morphology of C-S-H cementation and the wrapping of coal gangue, we use the above pictures, and refer to the following pictures for the same results. In these pictures, the wrapped objects with relative smoothness face and obvious edges are considered as gangue particles. Because no other substance with similar irregular shape can be found in the product of hydration reaction.

Considering the form of dense network formed by C-S-H cementation and abundant production (Figure 2-Figure 5), they form a package for coal gangue, even completely cover the gangue particles, and the boundary can not be accurately identified.

Figure 2. C-S-H gel & Gangue (Incomplete coverage)

Figure 3. C-S-H gel & Gangue (Completely coverage with uneven thickness)

Figure 4. C-S-H gel & Gangue (Complete coverage)

Figure 5. Floc structure in block

Similar observations are made in following papers.

(1) Zhu, C.; Zhou, N.; Guo, Y.; Li, M.; Cheng, Q. Effect of Doped Glass Fibers on Tensile and Shear Strengths and Microstructure of the Modified Shotcrete Material: An Experimental Study and a Simplified 2D Model. Minerals 2021, 11, 1053. https://doi.org/10.3390/min11101053

(2) Zhao, Y.; Taheri, A.; Karakus, M.; Deng, A.; Guo, L. The Effect of Curing under Applied Stress on the Mechanical Performance of Cement Paste Backfill. Minerals 2021, 11, 1107. https://doi.org/10.3390/min11101107

Based on the above considerations, we hope that the reviewers can agree with us.

â‘¡The X-ray results of the filling body show the content of mixed salt, which does not exist in the gangue particles (Table 2). That means a series of ion-exchange occurring.

Table 2. Chemical Components of Coal Gangue.

Chemical Composition

SiO2

Fe2O3

PbO

Al2O3

CaO

MgO

P2O5

K2O

S

LOI

Content

/%

49.22

6.21

0.73

22.96

4.88

0.78

0.12

1.2

0.8

13.1

â‘¢I am sorry for the wrong informations.

The Pb2+ is physically adsorbed by the negatively charged C-S-H.

And the equation(1) & (2) revised as following :

Pb2++ Ca(C-S-H)2→Pb(C-S-H)2+Ca2+  (1)

Pb2++2(C-S-H)-→Pb-(C-S-H)2  (2)

The equation (3) need to be revised to keep the charge balance:

3Pb2++7OH-+H++Ca2++ SO2- 4→Pb2SO4(OH)2·Pb(OH)2·Ca(OH)2·H2O     (3)

As the reviewer says, calcium ions and sulfate ions cannot be present in the aqueous phase at the same time because calcium sulfate will precipitate immediately.

The reasons for small amount CaSO4 in the solution are:

compared with calcium ion, the binding ability of lead ion and sulfate ion is higher; the following chemical reactions exist.

CO32-+CaSO4=CaCO3+SO42-

Point 2:  Some key information about the leaching tests is not given in this manuscript. For example, what is the solid/liquid ratio? What is the pH of leachate? How did the authors collect the liquid sample? Any mixing applied during the leaching tests? What is the device used for the leaching tests? Without these parameters, it is hard to understand the results shown in Figs. 8 and 9.

Response 2: I am sorry for the incomplete provide of information.

Page 7, line 220:

The solid/liquid ratio is 1:5.

The PH of the solution changes from 4.5 to 5.5 versus time. The changing acid environment will impact the strength backfill body and heavy metal leaching, however, it was not considered because the experiment was carried out in accordance with the actual engineering conditions. The in-depth research will be carried out in the future. I hope the train of thought could get the reviewers’ affirmation and support.

Page 7, line 219- line 222:

After discussion carefully, we decided to use the following diagram to deduce the experimentals and the collection of liquid samples. I hope Figure 8 will get the affirmation and support from reviewers.

Figure 8. Schematic diagram of leaching tests

The equipment used for the leaching tests is Inductively Coupled Plasma Spectrometer shows as following (Figure 2. in original paper).

Figure 2. Inductively Coupled Plasma Spectrometer

Reviewer 3 Report

The manuscript "Study on the Pb2+ Consolidation Mechanism of Gangue-Based Cemented Backfill" by Hao Wang, Qi Wang, Yuxin Hao and Yingying Wang was submitted for peer review.

I read the submitted manuscript with great interest. The authors turned to a very urgent problem: creating a backfill based on coal waste and studying its strength characteristics after hardening. 

The use of coal mining waste in backfill reduces the load on the environment by involving industrial waste into a closed production cycle.

The authors propose to use industrial waste in a material composition. To minimize the impact of industrial waste after using it as a component of backfill, the authors study on the Pb2+ Consolidation Mechanism.

The manuscript has significant flaws that need to be corrected. Correction of the shortcomings listed below must be done to improve the quality of the manuscript, enhance the ease of perception of the presented material and increase the interest of a readers.

1.) From my point of view, there are very few keywords. In addition, the keywords should be more direct and related to the content of the manuscript. Keywords enable the reader to quickly search for the necessary material and enable the author to popularize their research and increase interest and citations. But if this number of keywords satisfies the requirement of the journal, this comment is advisory.

2.) The abstract is not quite formed correctly. It is very blurry and framed incorrectly. It seems that the authors have taken certain phrases from the text and thus formed the abstract. The abstract should clearly indicate the purpose of the study, its importance for society (i.e. to characterize the problem), identify the methods and materials of the study, and the conclusions should be clearly and briefly formulated. There is no "starting point" in the abstract, that is, information about previous studies (one sentence is enough). From my point of view, in the abstract, such information begins with the statement: "Previously conducted studies have established that ...".

2.1) It is desirable to avoid narrative text in the abstract.

2.2) Try to use words and phrases: an analysis has been carried out; studied; developed; proposed; established and so on. It is advisable to start sentences in the abstract with these words and phrases.

2.3) At the end of the abstract, it is necessary to indicate the final result obtained by the authors, for example: A model has been developed that allows ...; A dependence has been established which is...; A pattern has been revealed...; An efficient system (technology) has been proposed, and so on.

The abstract should be revised.

3.) The manuscript has a sufficient list of references (57 references in total). But there is no comprehensive coverage of research in terms of geography of citations. There are no references to international experience in the field, especially to the work of East European, Ukrainian and Russian scientists. The list of references is intended to demonstrate the depth of the authors' study, the relevance of the material and interest of their research. 

3.1.) The depth of study is demonstrated with the number of references – is sufficient.

3.2.) Relevance – with the availability of research in recent years – is sufficient.

3.3.) Interest – with the availability of research by scientists from different countries - is not sufficient (practically absent). 

Since you are publishing your manuscript in an international publication, it is necessary to demonstrate the international relevance and interest of this issue. This can be done by analyzing the studies of scientists from different countries. It is imperative to supplement the list of references with studies of scientists from different countries over the past 3-5 years to show geographical (general/global) interest and relevance.

The List of References needs to be completed.

4.) In the introduction when analyzing previous studies, the authors make inaccuracies or provide information that overloads the text and often their claims are not accompanied with evidence. It is important for readers to know the essence (main idea) of the research you are referring to when analyzing previous work. In the introduction, it is necessary to analyze the previously completed work and note what has been done, what are the shortcomings, and what has been done incorrectly. Such shortcomings are present throughout the introduction. Authors need to revise the introduction, adjust, and supplement their statements with evidence.

5.) From my point of view, the authors abuse the names of scientists when mentioning the study, for example: Zhang. A reference [13-15] is sufficient. If the reader is interested in the name of the researcher, then it is easy to refer to the references list. It is important for the reader to know the essence (main idea) of the disclosed issue, not the name of the researcher.

6.) At the end of the introduction, there is no brief conclusion of the analytical study of earlier papers. The authors did not summarize their analysis and did not identify unresolved issues. This conclusion should make it possible to characterize the actual question posed, the purpose of the study and the tasks to be solved to achieve this goal. For example: Analyzing the above, it can be noted that ... is a very topical issue. Therefore, the purpose of this study is ... and to achieve this, it is necessary to solve the following tasks: 1); 2); ... Such a conclusion allows the reader to understand the vector of the study, and the authors to correctly formulate the conclusions. It needs to be improved. 

7.) Considering the comments (3) and (4), I would like to note that the authors have very poorly disclosed the main subject of the study. The impact of waste on the environment is great, so the issues of reducing this impact are very relevant and scientists around the world are trying to minimise it. In recent years, a lot of work has been carried out on the study of a material created on the basis of enrichment tailings with the aim of their subsequent use in a closed, waste-free (low-waste) production or for products intended for civil engineering. 

For example,

7.1) Kongar-Syuryun, Ch.; Aleksakhin, A.; Khayrutdinov, A.; Tyulyaeva, Y. Research of rheological characteristics of the mixture as a way to create a new backfill material with specified characteristics. Materials Today: Proceedings 2021, 38, 2052-2054. https://doi.org/10.1016/j.matpr.2020.10.139.

In this study, the authors study a material based on water-soluble ores tailings. Particular attention is paid to the material transportability with the maintaining the specified strength characteristics. Since the backfill is to be transported to the place of laying, it is necessary to pay attention to rheological characteristics or analyze previously performed work in this area.

7.2) Ermolovich, E.A.; Ivannikov, A.L.; Khayrutdinov, M.M.; Kongar-Syuryun, C.B.; Tyulyaeva, Y.S. Creation of a Nanomodified Backfill Based on the Waste from Enrichment of Water-Soluble Ores. Materials 2022, 15(10), 3689. https://doi.org/10.3390/ma15103689. 

This work is similar in terms of goals, objectives and research methods to the manuscript submitted for peer review. The authors study a composite based on waste from the processing of water-soluble ores. To increase the strength characteristics of the created material, fullerene-astarlene is used as a nanomodified additive. From my point of view, this work should be used in the analysis of previously performed studies, since it uses the methods of mechanical, microstructural, X-ray phase and petrographic analyzes to confirm arguments. 

7.3) Kongar-Syuryun, Ch.B.; Faradzhov, V.V.; Tyulyaeva, Yu.S.; Khayrutdinov, A.M. Effect of activating treatment of halite flotation waste in backfill mixture preparation. Mining Informational and Analytical Bulletin 2021, 2021(1), 43–57. https://doi.org/10.25018/0236-1493-2021-1-0-43-57.

7.4) Khayrutdinov, A.; Kongar-Syuryun, Ch.; Kowalik, T.; Faradzhov, V. Improvement of the backfilling characteristics by activation of halite enrichment waste for non-waste geotechnology. IOP Conf. Ser.: Mater. Sci. Eng. 2020, 867(1), 012018. https://doi.org/10.1088/1757-899X/867/1/012018.

In studies (7.3), (7.4), the authors propose the activation treatment of tailings before mixing to improve the strength and rheological characteristics. Activation treatment of the components or the addition of some kind of activating additive is one of the ways to improve the quality of the created material.

From my point of view, the works (7.3) and (7.4) will suit the authors in the analysis of previously completed works to demonstrate various options for controlling the characteristics of the created composite.

If the authors become familiar with the works presented in (7.1), (7.2), (7.3), (7.4) they will be able to properly form the introduction, enrich their manuscript with international research by scientists from Poland, Czech Republic, Slovenia, Slovakia, Russia, Germany and demonstrate the depth of their material, as well as eliminate the remarks (3) and (4). 

8) Of particular interest to me, and I think readers as well, is SEM and X-ray analysis. From my point of view, it is necessary to indicate the equipment on which the research was carried out (brand / model), the databases used in the X-ray analysis. Readers need to know all this so that they can repeat the experiment.

9.) When describing an experiment to create a backfill, you must specify:

9.1) the brand of Portland cement and preferably the cement manufacturer that the authors used;

9.2) how was the convergence of the results achieved;

9.3) how was the homogenization (mixing) carried out; what is the mixing tool; what is the mixing velocity and time;

9.4.) what is the sequence of filling of the components;

9.5) it is also not clear how the homogeneity of the composition (thoroughness of mixing) was achieved, provided that the amount of some components in the composite is minimal;

9.6) what equipment was used to study samples for uniaxial compression;

9.7) how underground (mine) conditions were achieved during the hardening of backfill;

To eliminate remarks (8) – (9), I would recommend reading the work (7.2). The recommended paper is similar to the one submitted for peer review. The article (7.2) describes the methodology in sufficient details.

Summary: The manuscript is a finished research work. But the corrections are needed. The chosen research topic is relevant. From my point of view, the authors failed to present their research correctly and clearly, which reduced its value and worsened the ease of perception of the material presented. 

From my point of view, the manuscript cannot be published in the open press without correction in accordance with my suggestions. 

Author Response

Response to Reviewer 3 Comments

Point 1: From my point of view, there are very few keywords. In addition, the keywords should be more direct and related to the content of the manuscript. Keywords enable the reader to quickly search for the necessary material and enable the author to popularize their research and increase interest and citations. But if this number of keywords satisfies the requirement of the journal, this comment is advisory.

Response 1: Thanks for the reviewer’s suggestions.

We revised keywords to make the article more popular and increase the researcher's interest and citations.

Delete: consolidation mechanism

Add: physical encapsulation; ion exchange; ion adsorption; chemical reaction

Point 2: The abstract is not quite formed correctly. It is very blurry and framed incorrectly. It seems that the authors have taken certain phrases from the text and thus formed the abstract. The abstract should clearly indicate the purpose of the study, its importance for society (i.e. to characterize the problem), identify the methods and materials of the study, and the conclusions should be clearly and briefly formulated. There is no "starting point" in the abstract, that is, information about previous studies (one sentence is enough). From my point of view, in the abstract, such information begins with the statement: "Previously conducted studies have established that ...".

2.1) It is desirable to avoid narrative text in the abstract.

2.2) Try to use words and phrases: an analysis has been carried out; studied; developed; proposed; established and so on. It is advisable to start sentences in the abstract with these words and phrases.

2.3) At the end of the abstract, it is necessary to indicate the final result obtained by the authors, for example: A model has been developed that allows ...; A dependence has been established which is...; A pattern has been revealed...; An efficient system (technology) has been proposed, and so on.

The abstract should be revised.

Response 2: Thanks for the reviewer’s professional advises, that is very important to make the article rationality.

Page 2-Page 3, line 106-line109:

We add some sentences to make the make the article more integrity.

Previously conducted studies have established that gangue direct backfill pollutes the groundwater due to the presence of heavy metal ions. However, the pollution risk of gangue-based cemented backfill needs to be studied by considering the change in the gangue particle size during cemented backfill as well as the physical and chemical reaction of fine particles such as fly ash and cement during the consolidation process.

Page 2, line 110-line 115:

The purpose of this study is to estimate and reduce the disasters stemming from gangue cemented backfill mining. To achieve this, it is necessary to carry out the following tasks: 1) Select appropriate elements and track their leaching behavior; 2) Screen for particle size and prepare samples, ICP tests, strength tests, etc; 3) Conduct an appropriate theoretical analysis.

Point 3: The manuscript has a sufficient list of references (57 references in total). But there is no comprehensive coverage of research in terms of geography of citations. There are no references to international experience in the field, especially to the work of East European, Ukrainian and Russian scientists. The list of references is intended to demonstrate the depth of the authors' study, the relevance of the material and interest of their research. 

3.1.) The depth of study is demonstrated with the number of references – is sufficient.

3.2.) Relevance – with the availability of research in recent years – is sufficient.

3.3.) Interest – with the availability of research by scientists from different countries - is not sufficient (practically absent). 

Since you are publishing your manuscript in an international publication, it is necessary to demonstrate the international relevance and interest of this issue. This can be done by analyzing the studies of scientists from different countries. It is imperative to supplement the list of references with studies of scientists from different countries over the past 3-5 years to show geographical (general/global) interest and relevance.

The List of References needs to be completed.

Response 3: Thanks for the reviewers’ constructive and wonderful suggestions.

We added some references to increase the international relevance and interest of this issue. The references focus on the utilization of solid waste and the strength characteristics of the products which published in recent years.

Added references as following.

(10) Kongar-Syuryun, Ch.; Aleksakhin, A.; Khayrutdinov, A.; Tyulyaeva, Y. Research of rheological characteristics of the mixture as a way to create a new backfill material with specified characteristics. Materials Today: Proceedings 2021, 38, 2052-2054. https://doi.org/10.1016/j.matpr.2020.10.139

(11) Ermolovich, E.A.; Ivannikov, A.L.; Khayrutdinov, M.M.; Kongar-Syuryun, C.B.; Tyulyaeva, Y.S. Creation of a Nanomodified Backfill Based on the Waste from Enrichment of Water-Soluble Ores. Materials 2022, 15(10), 3689. https://doi.org/10.3390/ma15103689

(50) Kongar-Syuryun, Ch.B.; Faradzhov, V.V.; Tyulyaeva, Yu.S.; Khayrutdinov, A.M. Effect of activating treatment of halite flotation waste in backfill mixture preparation. Mining Informational and Analytical Bulletin 2021, 2021(1), 43–57. https://doi.org/10.25018/0236-1493-2021-1-0-43-57

(51) Khayrutdinov, A.; Kongar-Syuryun, Ch.; Kowalik, T.; Faradzhov, V. Improvement of the backfilling characteristics by activation of halite enrichment waste for non-waste geotechnology. IOP Conf. Ser.: Mater. Sci. Eng. 2020, 867(1), 012018. https://doi.org/10.1088/1757-899X/867/1/012018

Point 4: In the introduction when analyzing previous studies, the authors make inaccuracies or provide information that overloads the text and often their claims are not accompanied with evidence. It is important for readers to know the essence (main idea) of the research you are referring to when analyzing previous work. In the introduction, it is necessary to analyze the previously completed work and note what has been done, what are the shortcomings, and what has been done incorrectly. Such shortcomings are present throughout the introduction. Authors need to revise the introduction, adjust, and supplement their statements with evidence.

Response 4: The reviewer’s suggestions are constructive and wonderful, we are deeply understand and support the views.

Based on the consideration of accuracy and preciseness, some sentences (Page 1, line 52-line 56) which insufficient to support the main idea of the research (include the original references[13][15][16][17][52]) were deleted.

Page 3, line 101:

The sentence “The cemented backfill … characteristics” was deleted, also the original reference [52].

Page 3, line 102-line 107:

The sentence revised as “The greater the degree of the damage to the cemented backfill body damage, the more de-veloped the internal fractures and the more evident the connection between the fractures, including the formation of network fractures.”

Page 2, line 63-line 66:

The sentence revised as “Qi [21], Xu [22], and Song [23,24] studied the relationship between the leaching strength of heavy metal elements from backfill body and the concentration of those elements in an external solution.”

We add the following sentences as a supplement at the end of introduction.

“Previously conducted studies have established that gangue direct backfill will polluted the groundwater due to the heavy metal ions. However, considering the change of particle size of gangue during cemented filling, as well as the physical and chemical reaction of fine particles such as fly ash and cement during the consolidation process, the pollution risk of gangue cemented backfill needs to be studied.”

Point 5: From my point of view, the authors abuse the names of scientists when mentioning the study, for example: Zhang. A reference [13-15] is sufficient. If the reader is interested in the name of the researcher, then it is easy to refer to the references list. It is important for the reader to know the essence (main idea) of the disclosed issue, not the name of the researcher.

Response 5: The reviewer’s view are very helpful for the preciseness of the article.

Page2 line 52-line 56:

We have deleted some similar references in the original article(original references [13-16]) for the remaining relevant documents are sufficient to illustrate the researcher's research views or timeliness.

Page2 line 57-line 60:

We delete the sentences for similar meaning, also the references ([18][19)).

Point 6: At the end of the introduction, there is no brief conclusion of the analytical study of earlier papers. The authors did not summarize their analysis and did not identify unresolved issues. This conclusion should make it possible to characterize the actual question posed, the purpose of the study and the tasks to be solved to achieve this goal. For example: Analyzing the above, it can be noted that ... is a very topical issue. Therefore, the purpose of this study is ... and to achieve this, it is necessary to solve the following tasks: 1); 2); ... Such a conclusion allows the reader to understand the vector of the study, and the authors to correctly formulate the conclusions. It needs to be improved. 

Response 6: The question has been answered in “Resopnse 2”.

Point 7: Considering the comments (3) and (4), I would like to note that the authors have very poorly disclosed the main subject of the study. The impact of waste on the environment is great, so the issues of reducing this impact are very relevant and scientists around the world are trying to minimise it. In recent years, a lot of work has been carried out on the study of a material created on the basis of enrichment tailings with the aim of their subsequent use in a closed, waste-free (low-waste) production or for products intended for civil engineering. 

For example,

7.1)

In this study, the authors study a material based on water-soluble ores tailings. Particular attention is paid to the material transportability with the maintaining the specified strength characteristics. Since the backfill is to be transported to the place of laying, it is necessary to pay attention to rheological characteristics or analyze previously performed work in this area.

7.2) Ermolovich, E.A.; Ivannikov, A.L.; Khayrutdinov, M.M.; Kongar-Syuryun, C.B.; Tyulyaeva, Y.S. Creation of a Nanomodified Backfill Based on the Waste from Enrichment of Water-Soluble Ores. Materials 2022, 15(10), 3689. https://doi.org/10.3390/ma15103689. 

This work is similar in terms of goals, objectives and research methods to the manuscript submitted for peer review. The authors study a composite based on waste from the processing of water-soluble ores. To increase the strength characteristics of the created material, fullerene-astarlene is used as a nanomodified additive. From my point of view, this work should be used in the analysis of previously performed studies, since it uses the methods of mechanical, microstructural, X-ray phase and petrographic analyzes to confirm arguments. 

7.3) Kongar-Syuryun, Ch.B.; Faradzhov, V.V.; Tyulyaeva, Yu.S.; Khayrutdinov, A.M. Effect of activating treatment of halite flotation waste in backfill mixture preparation. Mining Informational and Analytical Bulletin 2021, 2021(1), 43–57. https://doi.org/10.25018/0236-1493-2021-1-0-43-57.

7.4) Khayrutdinov, A.; Kongar-Syuryun, Ch.; Kowalik, T.; Faradzhov, V. Improvement of the backfilling characteristics by activation of halite enrichment waste for non-waste geotechnology. IOP Conf. Ser.: Mater. Sci. Eng. 2020, 867(1), 012018. https://doi.org/10.1088/1757-899X/867/1/012018.

In studies (7.3), (7.4), the authors propose the activation treatment of tailings before mixing to improve the strength and rheological characteristics. Activation treatment of the components or the addition of some kind of activating additive is one of the ways to improve the quality of the created material.

From my point of view, the works (7.3) and (7.4) will suit the authors in the analysis of previously completed works to demonstrate various options for controlling the characteristics of the created composite.

If the authors become familiar with the works presented in (7.1), (7.2), (7.3), (7.4) they will be able to properly form the introduction, enrich their manuscript with international research by scientists from Poland, Czech Republic, Slovenia, Slovakia, Russia, Germany and demonstrate the depth of their material, as well as eliminate the remarks (3) and (4). 

Response 7: Thanks for the reviewer's constructive and professional suggestions. We revised the abstract by add some sentences and references.

Page 1, line 42-line 46:

Reducing the impact of waste on the environment is a relevant issue, and scientists around the world are trying to minimize it. In recent years, a great amount of work has been carried to study a material created on the basis of enrichment tailings with the aim of their subsequent use in a closed, waste-free (low-waste) production or in products intended for civil engineering [10,11].

Point 8: Of particular interest to me, and I think readers as well, is SEM and X-ray analysis. From my point of view, it is necessary to indicate the equipment on which the research was carried out (brand / model), the databases used in the X-ray analysis. Readers need to know all this so that they can repeat the experiment.

Response 8: The reviewer's suggestion is professional and have constructive meaning for the integrity of the article.

The X-ray information and the test setup are shown in Table 3.

Table 3. The information of the X-ray and the test setup

Name

D/Max-3B X-ray diffractometer

Manufacturer

Rigaku (Japanese)

Test conditions

Cu target, K radiation, graphite curved crystal monochromator

Slit system

DS (divergent slit):1°

RS (receiving slit):1°

SS (anti-scattering slit):0.15mm

RSM (monochromator slit): 0.6°

X-ray tube voltage

35kV

X-ray tube current

30mA

Qualitative analysis

Scanning mode: Continuous scanning;

Scanning speed: 3°/min;

Sampling interval: 0.02°.

Quantitative analysis

Scanning mode: Step scanning;

Scanning speed: 0.25°/min;

Sampling interval: 0.01°

Databases

the standard powder diffraction data provided by JCPDS-ICDD

Point 9: When describing an experiment to create a backfill, you must specify:

9.1) the brand of Portland cement and preferably the cement manufacturer that the authors used;

Response 9.1): I am sorry for the missing information of the cement.

Page 4, line 148:

And the brand of the Portland cement (42.5) is Tianrui which produced by Zhengzhou Tianrui Cement Co., Ltd.

9.2) how was the convergence of the results achieved;

Response 9.2): The reviewers' opinions are very professional.

Page 5, line 185:

Every experiment was carry out three times and take the average value to achieve low error.

9.3) how was the homogenization (mixing) carried out; what is the mixing tool; what is the mixing velocity and time;

Response 9.3):

Page 5, line 172-line 174:

The mixing equipment is Cement paste mixer.

Velocity and time setup:

     Revolution velocity: 62 r/min; Rotation velocity: 140 r/min; Time: 18 min.

9.4.) what is the sequence of filling of the components;

Response 9.4):

Page 5, line 174-line 177:

The filling sequence is:

Gangue particles (all)→Distilled water→Mixture of fly ash and cement (all).

During quickly stir mixing pay attention to prevent water splashing out.

9.5) it is also not clear how the homogeneity of the composition (thoroughness of mixing) was achieved, provided that the amount of some components in the composite is minimal;

Response 9.5): The method described in 7.2) is very professional, and have great significance to the integrity of this papers’ exposition.

Page5, line 177-line181:

Previously, the optimal amount of portland cement determined as 8% of the solid mass. Portland cement and fly ash were mixed for 3 min, then, the distilled water was added, and the mixing was continued for an additional 5 min until a homogeneous mass was achieved. Such a sequence of mixing is due to a sufficiently small amount of portland and will lead to its better distribution in the entire volume of the material being prepared.

9.6) what equipment was used to study samples for uniaxial compression;

Response 9.6):

Page 5, line 183-line 184:

The uniaxial compression test equipment is YAW-300E produced by YongCe Co., Ltd (China).

9.7) how underground (mine) conditions were achieved during the hardening of backfill;

Response 9.7):

Page 5, line 181-line 183:

The strength of backfill body is the decisive prerequisite for the implementation of backfill mining (T=20±2℃; W≥95%).

In this paper, we discussed the consolidation effect of cemented backfill on heavy metal elements in gangue, so as to reduce the risk of groundwater pollution. According to the tests and previous research, the uniaxial compressive strength of the backfill body increases with time and reaches more than 2MPa, which achieved the needs of underground (mine) conditions.

To eliminate remarks (8) – (9), I would recommend reading the work (7.2). The recommended paper is similar to the one submitted for peer review. The article (7.2) describes the methodology in sufficient details.

Round 2

Reviewer 1 Report

There is a lot of improvement in the manuscript.

Still there is a concern about the leaching test. Only leaching test figure is not enough to understand the whole procedure. Description along with figure is appropriate to present your work. The authors need add a detail description on the leaching test with reference of the test procedure.

Reviewer 2 Report

The Reviewer appreciates the changes made by the Authors. However, the current version of manuscript still needs improvements before it can be considered for publication. The Reviewer raises the following points for Authors’ consideration for revision.

1. The Reviewer appreciates the explanation and relating references about the formation of C-S-H on the gangue particle surface. It seems that the gangue particle surface is smooth, while the C-S-H gel has a rough surface texture. Is this the major difference between gangue and C-S-H gel under the SEM?

2. What is the composition of the mixed salt shown in the XRD results in Figure 11? Also, the formation of mixed salt can be due to the adsorption of Pb2+ on the C-S-H gel. It does not necessarily mean the ion-exchange occurs.

3.  Lines 269-270: The Reviewer is still confused with the adsorption mechanism of the Pb2+ ions by the C-S-H gel. The Authors first mentioned that Pb2+ should be physically adsorbed by the negatively charged C-S-H, but then stated that “the chemical adsorption of Pb2+ by the C-S-H gel is usually shown in Pb2+ bonds with Ca2+ and Si4+ and forms precipitates.” What is the adsorption type for Pb2+ on C-S-H? Physical or chemical? If it is the former, the Authors should explain how the C-S-H obtains negative charges in the aqueous phase? If the latter is the case, chemical reactions should be shown.

Reviewer 3 Report

The manuscript "Study on the Pb2+ Consolidation Mechanism of Gangue-Based Cemented Backfill" by Hao Wang, Qi Wang, Yuxin Hao, Yingying Wang and Burui Ta was submitted for second review. 

As can be seen from the submitted manuscript and the explanatory note to the review, the authors did a lot of work to make changes in accordance with the comments.

The revised manuscript is a completed scientific study on a highly relevant topic: creating a backfill based on coal waste and studying its strength characteristics after hardening. The revised version of the manuscript, in my opinion, fully satisfies the requirements of a scientific article and can be published in the open press. 
